# Effects of intravenous magnesium sulfate on serum calcium-regulating hormones and plasma and urinary electrolytes in healthy horses

**Stephen A. Schumacher**[1,2], **Ahmed M. Kamr**[1,3], **Jeffrey Lakritz**[1], **Teresa A. Burns**[1], **Alicia L. Bertone**[1], **Ramiro E. Toribio**[1]*

1 College of Veterinary Medicine, The Ohio State University, Columbus, OH, United States of America,
2 United States Equestrian Federation, Columbus, Ohio, United States of America, 3 Faculty of Veterinary Medicine, University of Sadat City, Sadat City, Egypt

* toribio.1@osu.edu

**Data Availability Statement:** Data has been deposited in the Open Science Framework: doi.org/10.17605/osf.io/ckvbh.

## Abstract

Intravenous magnesium sulfate ($MgSO_4$) is used in equine practice to treat hypomagnesemia, dysrhythmias, neurological disorders, and calcium dysregulation. $MgSO_4$ is also used as a calming agent in equestrian events. Hypercalcemia affects calcium-regulating hormones, as well as plasma and urinary electrolytes; however, the effect of hypermagnesemia on these variables is unknown. The goal of this study was to investigate the effect of hypermagnesemia on blood parathyroid hormone (PTH), calcitonin (CT), ionized calcium ($Ca^{2+}$), ionized magnesium ($Mg^{2+}$), sodium ($Na^+$), potassium ($K^+$), chloride ($Cl^-$) and their urinary fractional excretion (F) after intravenous administration of $MgSO_4$ in healthy horses. Twelve healthy female horses of 4–18 years of age and 432–600 kg of body weight received a single intravenous dose of $MgSO_4$ (60 mg/kg) over 5 minutes, and blood and urine samples were collected at different time points over 360 minutes. Plasma $Mg^{2+}$ concentrations increased 3.7-fold over baseline values at 5 minutes and remained elevated for 120 minutes ($P < 0.05$), $Ca^{2+}$ concentrations decreased from 30–60 minutes ($P < 0.05$), but $Na^+$, $K^+$ and $Cl^-$ concentrations did not change. Serum PTH concentrations dropped initially to rebound and remain elevated from 30 to 60 minutes, while CT concentrations increased at 5 minutes to return to baseline by 10 minutes ($P < 0.05$). The FMg, FCa, FNa, FK, and FCl increased, while urine osmolality decreased from 30–60 minutes compared baseline ($P < 0.05$). Short-term experimental hypermagnesemia alters calcium-regulating hormones (PTH, CT), reduces plasma $Ca^{2+}$ concentrations, and increases the urinary excretion of $Mg^{2+}$, $Ca^{2+}$, $K^+$, $Na^+$ and $Cl^-$ in healthy horses. This information has clinical implications for the short-term effects of hypermagnesemia on calcium-regulation, electrolytes, and neuromuscular activity, in particular with increasing use of Mg salts to treat horses with various acute and chronic conditions as well as a calming agent in equestrian events.

**Funding:** This study was funded by the United States Equestrian Federation (USEF) and the Ohio State University, which also provided support in the form of salary (SS), but did not have a role in study design, data collection and analysis, decision to publish, or preparation of the manuscript.

**Competing interests:** Dr. Stephen Schumacher is employed by the sponsor of the project (USEF). However, this affiliation does not alter our adherence to all PLOS ONE policies on sharing data and materials.

## Introduction

Intravenous magnesium sulfate ($MgSO_4$) is used in equine medicine to treat dysrhythmias, neurological disorders, hypomagnesemia in critical illness, and refractory hypocalcemia [1]. More recently, intravenous $MgSO_4$ has been used in equine events as a calming and performance-enhancing agent, with the ultimate goal of giving a competitive edge to horses that receive it [2,3].

Magnesium (Mg) is an essential macroelement involved in a multitude of physiological processes, including enzymatic activation, intermediary metabolism of carbohydrates, fats, and proteins, nucleic acid synthesis, cell membrane ion transport, neuromuscular excitability, cell proliferation, and calcium homeostasis [1]. The secretion of PTH and activity of the PTH receptor are modulated by Mg. Thus, Mg depletion could impair calcium regulation [4].

Similar to total calcium (Ca), in which ionized calcium ($Ca^{2+}$) represents the active fraction, ionized Mg ($Mg^{2+}$) is the active fraction of total Mg. Considering that a multitude of cellular processes are highly dependent on Mg, intracellular Mg concentrations are tightly regulated [1]. However, extracellular $Mg^{2+}$ is not under tight hormonal homeostatic control as is $Ca^{2+}$, and its plasma concentrations depend on gastrointestinal absorption, renal excretion, and bone exchange [1,5]. A number of hormonal and non-hormonal factors influence extracellular $Mg^{2+}$ concentrations [6].

Parathyroid hormone (PTH) and calcitonin (CT) are important Ca-regulating hormones [7,8]. Hypocalcemia stimulates and hypercalcemia suppresses PTH secretion, while the opposite occurs with CT release [9]. In addition to its effects on $Ca^{2+}$ homeostasis, PTH stimulates renal reabsorption of $Mg^{2+}$ and its release from bone [10]. Calcitonin (CT) is secreted by the thyroid gland C cells in response to hypercalcemia to restore normocalcemia by blocking osteoclast-mediated bone resorption and reducing renal calcium reabsorption. There is evidence in other species that hypermagnesemia also stimulates CT secretion [11,12].

The ability of the parathyroid gland chief cells and thyroid gland C cells to detect extracellular $Ca^{2+}$ concentrations depends on the calcium-sensing receptor (CaSR), a G-protein-coupled cell surface receptor [13,14]. The CaSR is also expressed in the renal tubular cells where it regulates the reabsorption of $Ca^{2+}$, $Mg^{2+}$, $Na^+$, $K^+$, $Cl^-$, and water in various species, including the horse [4,13,15]. Excessive CaSR activation promotes diuresis and could lead to dehydration in patients with chronic hypercalcemia [4,15]. Relevant to this study, the CaSR also detects other polyvalent cations, including $Mg^{2+}$, $Sr^{2+}$, $Gd^{3+}$, and neomycin [16]. Therefore, hypermagnesemia could potentially influence PTH and CT secretion, plasma $Ca^{2+}$ concentrations, as well as the urinary excretion of electrolytes.

Hypercalcemia affects calcium-regulating hormones, as well as plasma and urinary electrolytes in horses [4]. In addition, hypercalcemia decreases $Mg^{2+}$ concentrations in horses [4]. Therefore, based on the $Ca^{2+}$ and $Mg^{2+}$ interactions, as well as the ability of $Mg^{2+}$ to activate CaSR, one can speculate that hypermagnesemia could affect a number of blood and urinary variables in horses, which could have clinical implications, including calcium homeostasis, urinary electrolyte wasting, neuromuscular excitability, and cardiovascular function.

The goal of this study was to investigate the effect of experimentally induced hypermagnesemia with intravenous $MgSO_4$ on blood PTH, CT, $Mg^{2+}$, $Ca^{2+}$, $Na^+$, $K^+$, and $Cl^-$ concentrations, as well as on their urinary excretion in healthy horses. We hypothesized that hypermagnesemia will reduce $Ca^{2+}$ concentrations, initially suppress to later stimulate PTH secretion, with the opposite effects on CT concentrations. We also proposed that hypermagnesemia, similar to hypercalcemia, will increase the urinary fractional excretion of $Mg^{2+}$, $Ca^{2+}$, $Na^+$, $K^+$ and $Cl^-$ in horses.

## Materials and methods

### Animals criteria and experimental design

This study was approved by The Ohio State University Institutional Animal Care and Use Committee and adhered to the principles of humane treatment of animals in veterinary clinical investigations as stated by the American College of Veterinary Internal Medicine and National Institute of Health guidelines.

Twelve healthy Standardbred mares with a median age of 8 years (4–18 years) and weighing 509 kg (432–600 kg) from The Ohio State University teaching herd were included in the study. Horses had no history of illness 6 months prior to the study, were under a routine health program, and were considered healthy based on physical examination, hematology, serum chemistry, and urine analysis. They were fed the same diet of grass and grass hay (0.5% calcium and 0.25% phosphorus), alfalfa hay (1.4% calcium and 0.25% phosphorus), and no concentrate feed.

### MgSO$_4$ administration

All horses were administered a single intravenous dose of MgSO$_4$ (60 mg/kg) (Magnesium Sulfate 50% solution, Wedgewood Pharmacy, Swedesboro, NJ, USA) over 5 minutes to induce hypermagnesemia. This dose was chosen because it increases Mg$^{2+}$ concentrations 3-fold over baseline values, which is expected to activate CaSR, and is also the dose anecdotally used for its calming effects at equestrian events [3]. This represented approximately 30 grams of MgSO$_4$ (60 ml) for a 500 kg horse [3].

### Sampling

Polyurethane catheters (Mila International, Florence, KY, USA) were placed aseptically in the left and right jugular veins for MgSO$_4$ administration and blood sample collection and were removed after the last blood sample was collected. Blood samples were collected at 0 (baseline; before MgSO$_4$ administration), 5, 10, 15, 30, 45, 60, 120, 180, 240, 300, and 360 minutes after MgSO$_4$ administration. A Foley catheter was placed aseptically into the bladder and urine samples collected at baseline (time 0; before MgSO$_4$ administration) and at 10, 30, 60, 120, 180, 240, 300, and 360 minutes after MgSO$_4$ administration.

Blood was collected into serum clot tubes, allowed to clot for one hour at room temperature and centrifuged at 2,000 × g for 10 minutes at 4˚C. Serum was aliquoted in smaller volumes and stored at -80˚C until analysis. At each time point, blood was also collected into heparinized syringes for immediate electrolyte and creatinine measurements. Urine samples were processed immediately after collection to measure electrolyte and creatinine concentrations. The urinary fractional excretion of Mg$^{2+}$ (FMg), Ca$^{2+}$ (FCa), Na$^+$ (FNa), K$^+$ (FK), and Cl$^-$ (FCl) was calculated as [Ux/Sx]/[Ucr/Scr]×100, where U = urine, S = serum, x = each electrolyte, and cr = creatinine concentrations [4]. Urinary fractional excretions were calculated from 9 horses and results are expressed as a fraction (%) of the urinary excretion of creatinine [4].

### Laboratory analyses

Concentrations of Mg$^{2+}$, Ca$^{2+}$, Na$^+$, K$^+$, Cl$^-$, and creatinine were determined in heparinized plasma with a biochemistry analyzer (pHOx Ultra Analyzer, Nova Biomedical Corp., Waltham, MA, USA). Urine creatinine and electrolyte concentrations were measured using an automated chemistry system (Cobas C501, Roche Diagnostics, Indianapolis, IN, USA).

## PTH and calcitonin measurements

Serum PTH concentrations were measured with a human-specific immunoradiometric assay (Scantibodies Laboratory, Santee, CA, USA) with a working range of 6.5–2328 pg/mL, a sensitivity of 1 pg/mL, and previously validated for horses [17]. Serum CT concentrations were determined with a human-specific immunoradiometric assay (Scantibodies Laboratory, Santee, CA, USA) with a working range of 10–1000 pg/mL and a detection limit of 1 pg/mL. The CT assay was validated for equine samples, with intra and inter-assay coefficients of variation <10%, and linear parallelism at 1:2, 1:4 and 1:8 dilutions ($R^2 = 0.96$).

## Statistical analysis

Sample size was calculated based on a power of 0.8 and a significance of 0.05 (alpha) to demonstrate statistical PTH differences using information from a similar study on the effects of hypercalcemia on serum and urine electrolytes [4]. Based on this, 8 horses would suffice, but number was increased because a lower effect of Mg compared to calcium over calcium-regulating hormones and electrolytes was expected. Data were assessed for normality with Shapiro-Wilk statistic and were normally distributed. Therefore, results are expressed as means with standard errors (S.E.). Data were analyzed using repeated measures One-Way ANOVA with Tukey's test to determine significance between different time points (Prism 8.0, GraphPad Software, Inc., La Jolla, CA, USA). Figures were generated using graphics software (SigmaPlot 14, Systat Software, Inc., San Jose, CA, USA). Significance was set at $P < 0.05$.

# Results

## Plasma $Mg^{2+}$, $Ca^{2+}$, $Na^+$, $K^+$, and $Cl^-$ concentrations and $Mg^{2+}/Ca^{2+}$

Plasma $Mg^{2+}$ concentrations increased sharply by 5 minutes and remained significantly higher than baseline for 180 minutes (**Fig 1A** and **Table 1**; **P < 0.05**). At 5, 15 and 60 minutes, plasma $Mg^{2+}$ concentrations were 3.7, 2.8, and 1.7-fold baseline values, respectively ($P < 0.05$). At 360 minutes, 4 horses still had $Mg^{2+}$ concentrations that were 30% higher than their baseline values. Hypermagnesemia led to a sharp and short lasting increase in plasma $Ca^{2+}$ concentrations to then cause a steady and statistically significant decrease in $Ca^{2+}$ concentrations ($P < 0.05$), reaching its lowest values at 45 minutes (12.5% decrease from time 0) to subsequently return to baseline values at 120 minutes (**Fig 1A** and **Table 1**). Plasma $Na^+$, $K^+$, $Cl^-$, and creatinine concentrations and plasma osmolality did not change over time (**Table 1**). The $Mg^{2+}/Ca^{2+}$ ratio was elevated by 5 minutes and remained above baseline values for 120 minutes (**Table 1**; **P < 0.05**).

## PTH and CT concentrations

Serum PTH concentrations initially decreased followed by a marked increase at 30–60 minutes after $MgSO_4$ administration ($P < 0.05$), remaining elevated for the rest of the study although not statistically different than baseline (**Fig 1B** and **Table 1**). Serum CT concentrations increased rapidly at 5 minutes corresponding with the end of $MgSO_4$ administration and highest $Mg^{2+}$ concentrations ($P < 0.05$), decreasing at 10 minutes and remaining at baseline values for the rest of the study (**Fig 1B** and **Table 1**).

## Urinary electrolytes and osmolality

Urine $Mg^{2+}$ concentrations increased steadily to reach statistical significance at 120 and 180 minutes compared to time 0 (**Table 2**; **P < 0.05**), to decrease to baseline values by 240 minutes. Urinary $Ca^{2+}$ concentrations also increased by 180 minutes (**Table 2**; **P < 0.05**), to return to

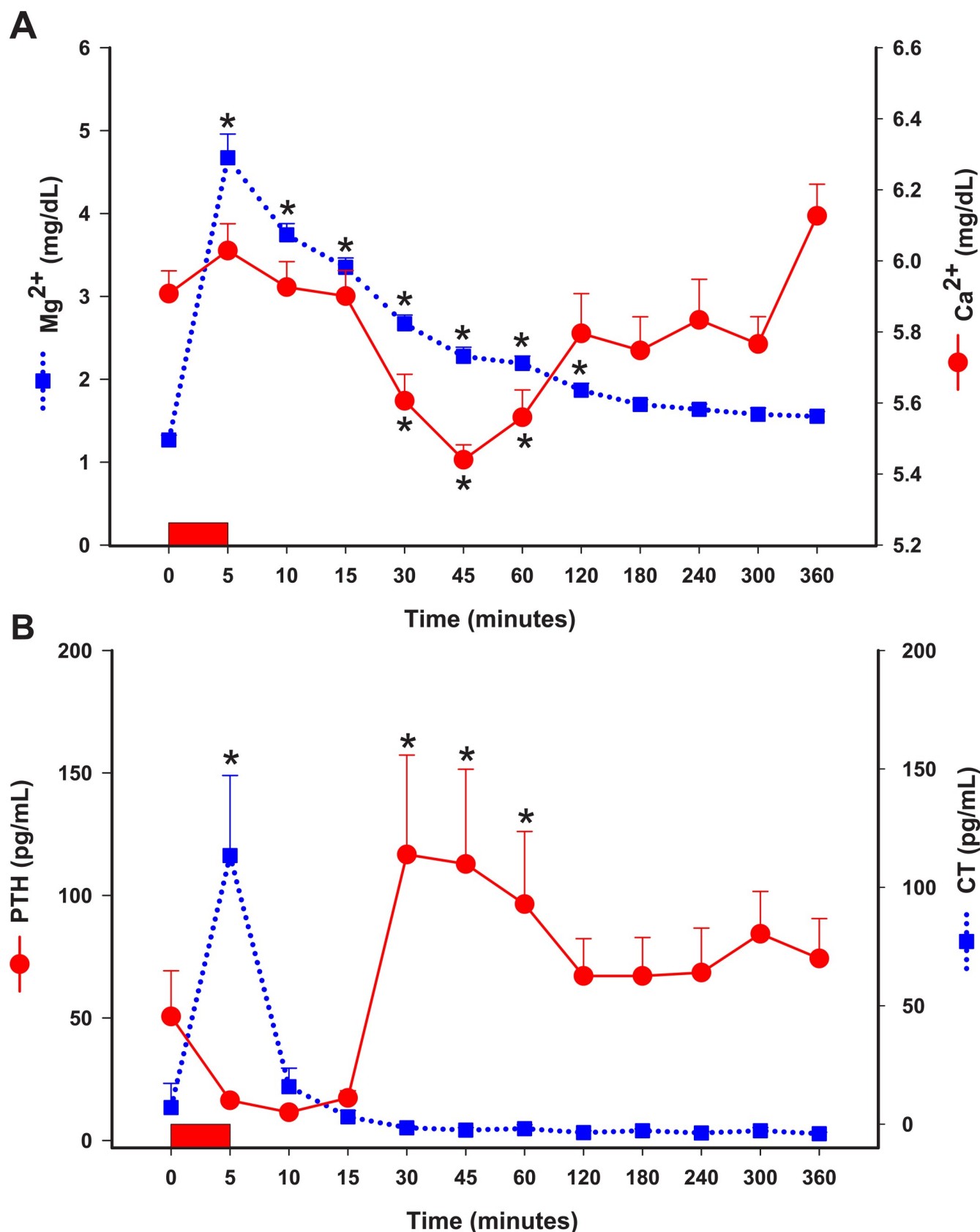

**Fig 1.** Plasma $Mg^{2+}$ and $Ca^{2+}$ **(A)** and serum PTH and CT **(B)** concentrations following IV administration of $MgSO_4$ (60 mg/kg) in healthy horses. Values expressed as mean with S.E. * indicates statistically different than time 0. Rectangle over X axis represents IV administration of $MgSO_4$ over 5 minutes. $Mg^{2+}$, plasma ionized magnesium; $Ca^{2+}$, plasma ionized calcium PTH, parathyroid hormone; CT, calcitonin.

baseline at 240 minutes. Urine $Na^+$ concentrations increased at 60 minutes, while urine $K^+$ concentrations decreased at 30 minutes and returned to baseline at 360 minutes (**Table 2**; **P < 0.05**). Urinary $Cl^-$ concentrations did not change overtime. Urinary osmolality decreased at 30 and 60 minutes to return to baseline thereafter (**Table 2**; **P < 0.05**).

### Urinary fractional excretion of electrolytes

The FMg increased by 3-fold at 30–120 minutes after $MgSO_4$ administration, then returned to baseline values by 360 minutes (**Fig 2A**; **Table 3**; **P < 0.05**). The FCa, FNa, FK, and FCl increased statistically by 60 minutes, and returned to baseline thereafter (**Fig 2A and 2B**; **Table 3**; **P < 0.05**).

### Discussion

In the present study, experimentally-induced hypermagnesemia in healthy horses led to a short lasting decrease in $Ca^{2+}$ concentrations, with transient changes in PTH and CT concentrations, increasing the urinary fractional excretion of $Mg^{2+}$, $Ca^{2+}$, $Na^+$, $K^+$, and $Cl^-$, and decreasing urine osmolality.

A single intravenous dose of $MgSO_4$ at 60 mg/kg resulted in sustained hypermagnesemia for over 120 minutes. Similar results were observed in horses in which $MgSO_4$ was evaluated as a treatment for trigeminal headshaking syndrome and in human patients with preeclampsia [18,19]. These findings also suggest that the in horses the physiological effects of the single dose of $MgSO_4$ used in this study could last up to 6 hours. In addition, this information may

**Table 1. Plasma $Mg^{2+}$, $Ca^{2+}$, $Na^+$, $K^+$ and $Cl^-$, and serum PTH and CT concentrations, and $Mg^{+2}/Ca^{2+}$ ratio following IV administration of $MgSO_4$ (60 mg/kg) in healthy horses.**

| Variables | Time (minutes) | | | | | | | | | | | |
|---|---|---|---|---|---|---|---|---|---|---|---|---|
| | 0 | 5 | 10 | 15 | 30 | 45 | 60 | 120 | 180 | 240 | 300 | 360 |
| $Mg^{2+}$ (mg/dL) | 1.3±0.1 | 4.7±0.3* | 3.7±0.1* | 3.3±0.1* | 2.7±0.1* | 2.3±0.1* | 2.2±0.1* | 1.9±0.1* | 1.7±0.1 | 1.6±0.1 | 1.6±0.1 | 1.6±0.1 |
| $Ca^{2+}$ (mg/dL) | 5.9±0.1 | 6.0±0.1 | 5.9±0.1 | 5.9±0.1 | 5.6±0.1* | 5.4±0.0* | 5.6±0.1* | 5.8±0.1 | 5.8±0.2 | 5.8±0.1 | 5.8±0.1 | 6.1±0.1 |
| $Mg^{2+}/Ca^{2+}$ | 0.2±0.0 | 0.8±0.1* | 0.6±0.0* | 0.6±0.0* | 0.5±0.0* | 0.4±0.0* | 0.4±0.0* | 0.4±0.0* | 0.3±0.0 | 0.3±0.0 | 0.3±0.0 | 0.3±0.0 |
| $Na^+$ (mmol/L) | 137.5±0.4 | 136.5±0.5 | 137.0±0.5 | 136.8±0.5 | 137.1±0.6 | 137.2±0.4 | 136.6±0.6 | 136.4±0.7 | 137.0±0.6 | 138.2±0.5 | 137.2±0.6 | 137.2±0.4 |
| $K^+$ (mmol/L) | 3.6±0.3 | 3.3±0.1 | 3.3±0.1 | 3.3±0.0 | 3.4±0.1 | 3.3±0.1 | 3.3±0.1 | 3.4±0.1 | 3.3±0.1 | 3.5±0.1 | 3.4±0.1 | 3.6±0.2 |
| $Cl^-$ (mmol/L) | 104.0±0.6 | 103.3±0.6 | 102.9±0.7 | 102.5±0.6 | 102.7±0.7 | 102.9±0.7 | 103.0±0.1 | 103.1±1.2 | 103.5±0.5 | 104.5±0.3 | 104.3±0.4 | 104.03±0.8 |
| PTH (pg/mL) | 45.5±19.3 | 10.1±1.4 | 5.0±1.1 | 11.0±2.7 | 113.9±37.6* | 109.8±40.0* | 92.8±30.7* | 62.6±15.7 | 62.6±16.2 | 64±18.8 | 80.3±17.9 | 69.9±16.8 |
| CT (pg/mL) | 13.4±9.9 | 116.3±32.6* | 22.0±7.5 | 9.7±2.3 | 5.1±1 | 4.2±0.1 | 4.8±1.7 | 3.2±0.8 | 4.0±0.8 | 3.0±0.7 | 4.0±1.2 | 2.7±0.7 |
| Creatinine (mg/dL) | 1.1±0.1 | 1.10±0.1 | 1.1±0.1 | 1.0±0.1 | 1.1±0.1 | 1.0±0.5 | 1.1±0.1 | 1.1±0.1 | 1.1±0.1 | 1.1±0.1 | 1.1±0.1 | 2.1±0.1 |
| Osmolality (mOsm/kg) | 275.9±1.5 | 274.6±1.7 | 275.1±1.5 | 275.3±1.3 | 276.0±1.6 | 276.8±1.8 | 275.2±1.8 | 275.0±1.2 | 275.6±1.3 | 272.8±4.5 | 275.2±1.5 | 275.3±1.8 |

$Mg^{2+}$, ionized magnesium; $Ca^{2+}$, ionized calcium; $Na^+$, sodium; $K^+$, potassium; $Cl^-$, chloride; PTH, parathyroid hormone; CT, calcitonin.

*$P<0.05$.

Values expressed as mean ± S.E.

**Table 2. Urine Mg$^{2+}$, Ca$^{2+}$, Na$^+$, K$^+$, Cl$^-$, and creatinine concentrations and urine osmolality following IV administration of MgSO$_4$ (60 mg/kg) in healthy horses.**

| Variables | Time (minutes) | | | | | | | | |
|---|---|---|---|---|---|---|---|---|---|
| | 0 | 10 | 30 | 60 | 120 | 180 | 240 | 300 | 360 |
| Mg$^{2+}$ (mg/dL) | 49.6±11.8 | 109.9±27.7 | 143±36 | 111.8±20.5 | 138.9±20.6* | 154.2±22.1* | 125.3±20.3 | 98.0±17.4 | 96.7±8.5 |
| Ca$^{2+}$ (mg/dL) | 59.7±28.2 | 67.2±26.5 | 51.4±8.1 | 71.9±45.4 | 109.5±84.8 | 119.5±94. 7* | 64.3±31.5 | 19.9±8.7 | 88.5±44.0 |
| Na$^+$ (mEq/L) | 34.2±9.7 | 64.9±13.8 | 74.3±17.2 | 115.3±37.1* | 62.8±20.2 | 38.0±14.0 | 34.3±12.5 | 38.0±19.7 | 18.4±0.9 |
| K$^+$ (mEq/L) | 272.8±17.6 | 206.3±28.2 | 134.5±21.7* | 172.8±31.5 | 221.4±24.8 | 263±26.3 | 266.2±26.4 | 226±34.7 | 267.9±24.7 |
| Cl$^-$ (mEq/L) | 174.1±23.8 | 150.8±18.8 | 153.6±16.6 | 152.8±21.4 | 145.1±17.3 | 133.3±11.9 | 139.9±23.4 | 131.6±36.1 | 139±35.1 |
| Creatinine (mg/dL) | 316.1±53.6 | 204.5±60.2 | 97.7±27.8* | 92.9±19.2* | 183.7±39.6 | 287±54.78 | 336.5±65.2 | 412.9±117.1 | 463.6±132.9 |
| Osmolality (mOsm/kg) | 1249±125.9 | 856±143 | 624±75.8* | 674.2±74.0* | 845±116.2 | 1028±129.8 | 1053±138.8 | 1103±145.1 | 1292±148.6 |

Mg$^{2+}$, ionized magnesium; Ca$^{2+}$, ionized calcium; Na$^+$, sodium; K$^+$, potassium; Cl$^-$, chloride.

*P < 0.05.

Values expressed as mean and S.E.

be relevant to venues such as equestrian competitions, knowing that for at least 60 minutes after administration of this dose, serum [Mg$^{2+}$] will be twice baseline values, potentially explaining the calming effects of hypermagnesemia in horses [2,3], which has potential regulatory connotations. An increase in Mg$^{2+}$ concentrations coupled with a high Mg$^{2+}$/Ca$^{2+}$ ratio could be a potential biomarker for nefarious administration of MgSO$_4$ to competing horses [3,20]. It was also demonstrated that hypermagnesemia reduced headshaking behavior in horses [19]. These actions of Mg have been attributed to its ability to block N-methyl-D-aspartate (NMDA) ionotropic receptors and voltage-dependent calcium channels, reducing neuronal excitability [1,2,21,22]. The inhibitory effect of Mg$^{2+}$ over NMDA receptor activity has also been the rationale for its use in acute brain injury, nociception, anesthesia, and behavioral disorders [1–3,19,21].

A decrease in Ca$^{2+}$ concentrations after MgSO$_4$ infusion has been documented in human patients, horses and dogs [3,19,23–26] and could be attributed to excessive Mg$^{2+}$ activation of CaSR in the renal tubules, increasing the urinary excretion of Ca and Mg, as reported in human patients [23,26,27]. We anticipated this response based on a previous study where experimental hypercalcemia decreased Mg$^{2+}$ in healthy horses [4]. It is also possible that hypermagnesemia suppressed PTH secretion by activating CaSR [23], at least at early time points in the horses of this study, because subsequently there was a PTH rebound response that we attributed to the drop in Ca$^{2+}$ concentrations. One difference between this study and publications using MgSO$_4$ as a tocolytic agent was the rapid intravenous administration of MgSO$_4$ to these horses compared to how it is administered to pregnant women [28]. One study in healthy horses showed that hypercalcemia has a similar effect as hypermagnesemia, increasing the urinary excretion of Ca$^{2+}$, Mg$^{2+}$, Na$^+$, K$^+$, and Cl$^-$, increasing urine output, and decreasing urine osmolality [4].

Parathyroid hormone promotes renal calcium reabsorption and bone resorption to maintain normocalcemia [29,30]. The PTH receptor, via cAMP, mediates the renal actions of PTH by increasing the activity of the Na$^+$/K$^+$/2Cl$^-$ type 2 cotransporter (NKCC2), raising the electropositivity of the renal tubular lumen to promote paracellular cation reabsorption [31]. Activation of CaSR interferes with NKCC2, similar to furosemide, and inhibits vasopressin-mediated traffic of aquaporin-containing vesicles, explaining the diuretic actions of hypercalcemia in multiple species [15,32,33]. Under a similar premise supported by physiological studies showing that Mg$^{2+}$ activates CaSR [13], hypermagnesemia could interfere with PTH secretion and renal reabsorption of electrolytes and water. In fact, CaSR has been described as

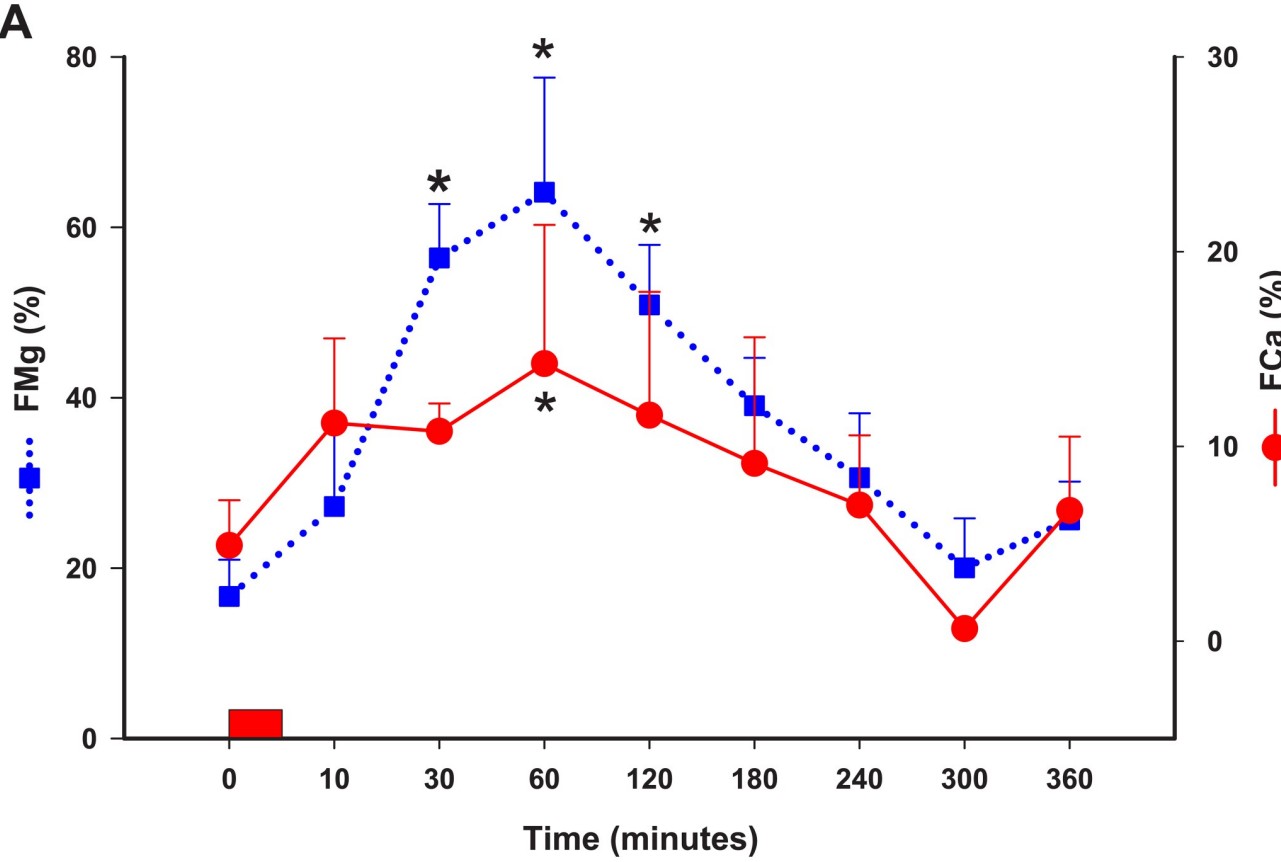

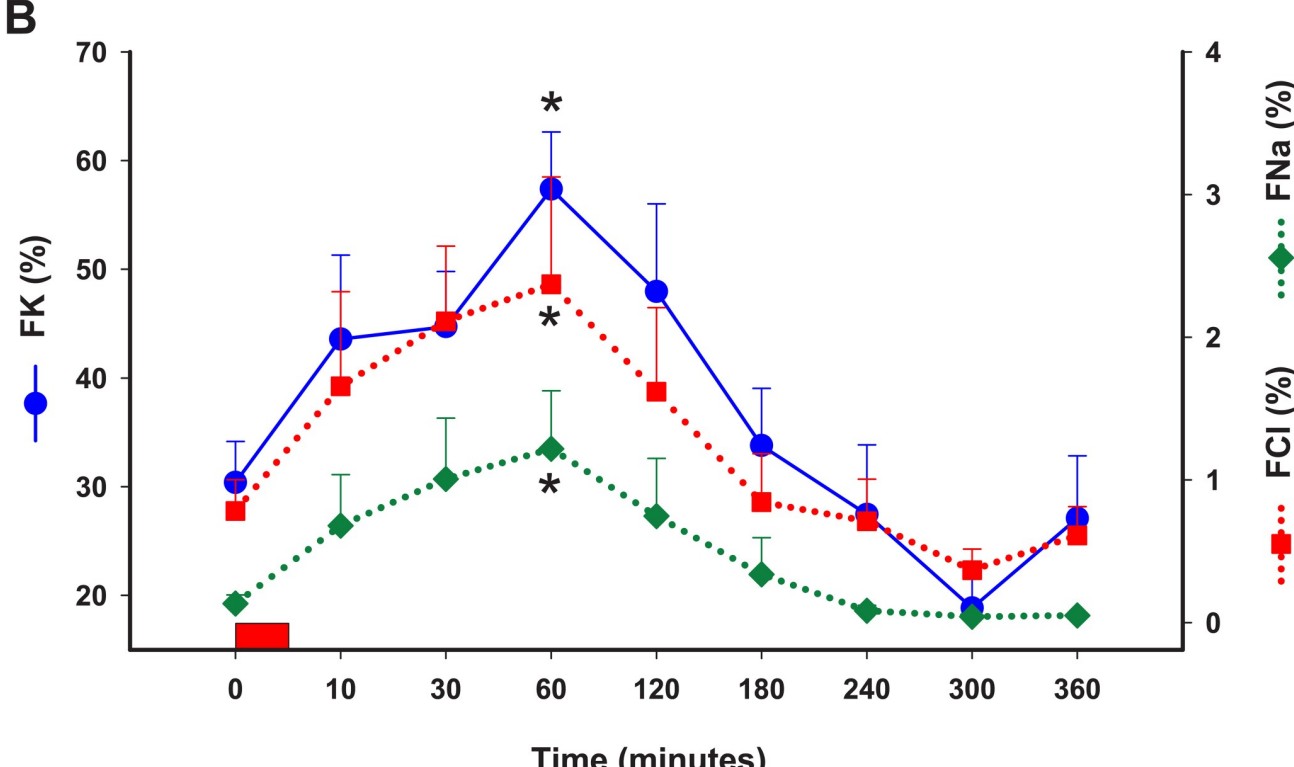

**Fig 2.** FMg and FCa **(A)** and FNa, FK and FCl **(B)** (%) following IV administration of MgSO$_4$ (60 mg/kg) in healthy horses. Values expressed as mean with S.E. * indicates statistically different than time 0 (P < 0.05). Rectangle over X axis represents IV administration of MgSO$_4$ over 5 minutes. FMg, urinary fractional excretion of Mg$^{2+}$; FCa, urinary fractional excretion of Ca$^{2+}$; FNa, urinary fractional excretion of Na$^+$; FK, urinary fractional excretion of K$^+$; FCl, urinary fractional excretion of Cl$^-$.

a promiscuous receptor because it can sense multiple cations at different extracellular concentrations [13].

The initial PTH suppression could be attributed to excessive CaSR activation by Mg$^{2+}$. This effect was likely counteracted by the drop in Ca$^{2+}$ concentrations, which likely triggered a rebound PTH response, reflecting the preference of CaSR for Ca$^{2+}$ over Mg$^{2+}$ [13]. The increase in serum PTH concentrations occurred in parallel with the decrease in Ca$^{2+}$ concentrations, supporting the rapid response of this system to protect against hypocalcemia [34,35]. After its rapid rise, CT concentrations returned to baseline values for the rest of the study, despite hypermagnesemia. This indicates that Mg$^{2+}$ stimulated the secretion of CT stored in granules but was not sufficient to stimulate additional synthesis and secretion, in particular in the presence of low to normal Ca$^{2+}$ concentrations. It is likely that a mechanism similar to that which occurred in the parathyroid gland chief cells, but with opposite endocrine secretion, was involved in thyroid gland C cell CT secretion. A similar CT response was observed in pigs after magnesium chloride infusion [36].

In addition to the increased FMg and FCa, the most feasible explanation for increased FNa, FK, and FCl were the effects of increased extracellular Mg$^{2+}$ on transepithelial ion transport and water reabsorption, as previously discussed [37,38].

While no effect of hypermagnesemia on plasma Na$^+$, K$^+$, and Cl$^-$ concentrations was documented, its evident effect on urinary electrolyte excretion indicates that in the short term, other homeostatic systems could maintain their concentrations. Evaluating the effect of prolonged hypermagnesemia on calcium-regulating hormones and electrolyte balance could provide additional insight on equine Mg$^{2+}$ physiology.

The clinical implications of short-term hypermagnesemia on behavior, cardiovascular function, and neuromuscular activity in horses were recently documented [2,3,19], further demonstrating that Mg is a pleiotropic ion. Magnesium supplementation is recommended for a number of human disorders and is being promoted as a pre-anesthetic to reduce pain [21]. In addition, due to its calcium-channel and NMDA receptor blocking properties and prolonged half-life in horses, MgSO$_4$ supplementation should be considered for conditions associated with nociception, neuronal hyperexcitability, and energy dysregulation [1,2,21]. However, it is

**Table 3. Urinary fractional excretion of Mg$^{2+}$, Ca$^{2+}$, Na$^+$, K$^+$, and Cl$^-$ following IV administration of MgSO$_4$ (60 mg/kg, IV) in healthy horses.**

| Variables | Time (minutes) | | | | | | | | |
|---|---|---|---|---|---|---|---|---|---|
| | 0 | 10 | 30 | 60 | 120 | 180 | 240 | 300 | 360 |
| **FMg (%)** | 16.7±4.3 | 27.2±10.5 | 56.4±6.3* | 64.1±13.5* | 50.9±7.1* | 39.1±5.6 | 30.6±7.6 | 20.0±5.8 | 25.7±4.5 |
| **FCa (%)** | 4.9±2.3 | 11.2±4.4 | 10.8±1.4 | 14.3±7.1* | 11.6±6.4 | 9.1±6.5 | 7.0±3.6 | 0.7±0.2 | 6.7±3.8 |
| **FNa (%)** | 0.13±0.1 | 0.7±0.4 | 1.0±0.4 | 1.2±0.4* | 0.8±0.4 | 0.3±0.3 | 0.1±0.0 | 0.04±0.0 | 0.1±0.0 |
| **FK (%)** | 30.4±3.8 | 43.6±7.7 | 44.7±5.1 | 57.4±5.3* | 48.0±8.1 | 33.8±5.3 | 27.5±6.4 | 18.6±4.2 | 27.1±5.7 |
| **FCl (%)** | 0.8±0.2 | 1.7±0.7 | 2.1±0.5 | 2.4±0.8* | 1.6±0.6 | 0.9±0.3 | 0.7±0.3 | 0.4±0.2 | 0.6±0.2 |

FMg, urinary fractional excretion of magnesium; FCa, urinary fractional excretion of calcium; FNa, urinary fractional excretion of sodium; FK, urinary fractional excretion of potassium; FCl, urinary fractional excretion of chloride.

*P < 0.05.

Values expressed as mean ± S.E.

important to emphasize that intravenous administration of $MgSO_4$ to horses to give a competitive edge is a nefarious and dangerous practice.

The effect of intravenous $MgSO_4$ at the dose used in this study has not been prospectively evaluated in hospitalized horses, although lower doses of $MgSO_4$ are routinely used in equine patients with hypomagnesemia or dysrhythmias. It is anticipated that similar effects to the horses of this study would be expected in these patients. Oral magnesium supplements are used by horse owners and trainers to enhance systemic health and in general considered a low risk of causing hypermagnesemia because doses are low. One can speculate that chronic hypermagnesemia in horses could alter calcium and electrolyte homeostasis as well as neuromuscular activity, however, high Mg doses for a long time would be required. This would be a valid question to investigate to further understand Mg biology in healthy and sick horses.

## Conclusions

A single intravenous bolus administration of $MgSO_4$ to healthy horses causes prolonged hypermagnesemia, a transient decrease in $Ca^{2+}$ concentrations, and rapid changes in calcium-regulating hormones, increasing the urinary excretion of electrolytes and decreasing urine osmolality. This information enhances our understanding of equine Mg and Ca biology and has clinical implications. For example, based on information from this study, it would be of interest to know whether prolonged hypermagnesemia from excessive Mg supplementation leads to electrolyte depletion, water wasting, altered calcium homeostasis, with subsequent clinical consequences, including changes in neuromuscular activity, cardiovascular function, and behavior.

## Author Contributions

**Conceptualization:** Stephen A. Schumacher, Alicia L. Bertone, Ramiro E. Toribio.

**Data curation:** Stephen A. Schumacher, Ahmed M. Kamr, Ramiro E. Toribio.

**Formal analysis:** Stephen A. Schumacher, Ahmed M. Kamr.

**Funding acquisition:** Stephen A. Schumacher, Alicia L. Bertone, Ramiro E. Toribio.

**Investigation:** Stephen A. Schumacher, Ahmed M. Kamr, Jeffrey Lakritz, Teresa A. Burns, Alicia L. Bertone, Ramiro E. Toribio.

**Methodology:** Stephen A. Schumacher, Ahmed M. Kamr, Jeffrey Lakritz, Ramiro E. Toribio.

**Project administration:** Stephen A. Schumacher, Alicia L. Bertone, Ramiro E. Toribio.

**Resources:** Stephen A. Schumacher, Alicia L. Bertone, Ramiro E. Toribio.

**Software:** Stephen A. Schumacher, Ahmed M. Kamr.

**Supervision:** Alicia L. Bertone, Ramiro E. Toribio.

**Validation:** Ahmed M. Kamr, Ramiro E. Toribio.

**Visualization:** Stephen A. Schumacher, Ahmed M. Kamr.

**Writing – original draft:** Stephen A. Schumacher, Ahmed M. Kamr, Jeffrey Lakritz, Teresa A. Burns, Alicia L. Bertone, Ramiro E. Toribio.

**Writing – review & editing:** Stephen A. Schumacher, Ahmed M. Kamr, Jeffrey Lakritz, Teresa A. Burns, Alicia L. Bertone, Ramiro E. Toribio.

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
