## [Decision Letter · Decision Letter 0]

27 Mar 2021

PONE-D-21-04318

Effects of Intravenous Magnesium Sulfate on Calcium-Regulating Hormones, Plasma and Urinary Electrolytes in Healthy Horses

PLOS ONE

Dear Dr. Toribio,

Thank you for submitting your manuscript to PLOS ONE. After careful consideration, we feel that it has merit but does not fully meet PLOS ONE’s publication criteria as it currently stands. Therefore, we invite you to submit a revised version of the manuscript that addresses the points raised during the review process.

Both experts reviewers are very supportive of your manuscript but have highlighted some points that would need to be addressed through a minor revision.

We look forward to receiving your revised manuscript.

Kind regards,

Angel Abuelo, DVM, MRes, MSc, PhD, DABVP (Dairy), DECBHM

Academic Editor

PLOS ONE

Journal Requirements:

2. Thank you for stating the following in the Competing Interests:

"Dr. Stephen Schumacher is employed by the sponsor of the project (USEF). However, experimental design and studies were carried out by guidance of faculty from The Ohio State University."

We note that one or more of the authors have an affiliation to the commercial funders of this research study : United States Equestrian Federation.

2.1. Please provide an amended Funding Statement declaring this commercial affiliation, as well as a statement regarding the Role of Funders in your study. If the funding organization did not play a role in the study design, data collection and analysis, decision to publish, or preparation of the manuscript and only provided financial support in the form of authors' salaries and/or research materials, please review your statements relating to the author contributions, and ensure you have specifically and accurately indicated the role(s) that these authors had in your study. You can update author roles in the Author Contributions section of the online submission form.

2.2. Please also provide an updated Competing Interests Statement declaring this commercial affiliation along with any other relevant declarations relating to employment, consultancy, patents, products in development, or marketed products, etc. 

3. We noted in your submission details that a portion of your manuscript may have been presented or published elsewhere.

"Some clinical and laboratory information from a number of horses in this study were used in another publication"

Please clarify whether this publication was peer-reviewed and formally published. If this work was previously peer-reviewed and published, in the cover letter please provide the reason that this work does not constitute dual publication and should be included in the current manuscript.

5. Please remove your figures from within your manuscript file, leaving only the individual TIFF/EPS image files, uploaded separately.  These will be automatically included in the reviewers’ PDF.

Reviewers' comments:

Reviewer's Responses to Questions

**Comments to the Author**

1. Is the manuscript technically sound, and do the data support the conclusions?

Reviewer #1: Yes

Reviewer #2: Yes

2. Has the statistical analysis been performed appropriately and rigorously? 

Reviewer #1: I Don't Know

Reviewer #2: Yes

3. Have the authors made all data underlying the findings in their manuscript fully available?

Reviewer #1: Yes

Reviewer #2: Yes

4. Is the manuscript presented in an intelligible fashion and written in standard English?

Reviewer #1: Yes

Reviewer #2: Yes

5. Review Comments to the Author

Reviewer #1: This manuscript is well written, the physiology governing these electrolytes is systematically approached in the introduction and revisited in the discussion. The motivation for this study is clear to the reader; in order to understand how the nefarious use of supplements/drugs in competition horses might result in potential negative clinical implications we first need to understand their effect on physiological status of the horse. I would like to see a better explanation of how Mg has a calming influence on the demeanor of horses, if this is described, with citation(s).

Reviewer #2: This paper by Schumacher et al. investigates the effect of experimentally induced hypermagnesemia (using intravenous administration of magnesium sulfate) on blood parathyroid hormone (PTH), calcitonin, ionized calcium, ionized magnesium, sodium, potassium, chloride and their urinary fractional excretion. The authors speculate that the effects of hypermagnesemia on these plasma and urinary variables could have clinical implications and this forms the justification for the research.

Overall, this is a well considered and well conducted study. The manuscript is succinct and well written. The authors provide sound justification/reasoning for conducting the research. The introduction includes relevant information regarding the physiology of magnesium and calcium and the effects of magnesium on calcium regulation which provides appropriate background information for the study. Knowledge of the physiology also supports the hypothesis that there could be effects of hypermagnesemia on plasma and urinary variables. The authors clearly state their hypothesis at the end of the introduction.

The materials and methods section is succinct, yet provides adequate detail on how the study was conducted. The authors clearly explain how the dose of magnesium sulfate used in this study was determined.

The statistical analyses are appropriate and results are clearly presented. The figures provide a clear representation of the results and all results are provided in the accompanying tables.

The discussion provides a succinct summary of the results of the study and the authors clearly discuss their interpretation and provide excellent and detailed reasoning (based on physiological concepts and supported by relevant literature) for the results found. Ideas for research extending beyond this study, including investigating the effects of prolonged hypermagnesemia, are also discussed in the conclusion.

There are some minor points listed below which the authors should address before the submission can be accepted for publication.

1. Title: the title needs the addition of 'and' ("Effects of Intravenous Magnesium Sulfate on Calcium-Regulating Hormones and Plasma and Urinary Electrolytes in Healthy Horses') to clarify that the effects on plasma electrolytes etc is of interest, not the effects on plasma.

2. The research was conducted in 12 horses. Was a sample size calculation performed? This should be discussed in the materials and methods.

3. Line 78: At the end of this sentence, the authors state that the effects of hypermagnesemia could have clinical implications. The authors should expand on this slightly.

4. The authors conclude that the hypermagnesemia alters calcium-regulating hormones, reduces plasma calcium concentrations and increases the urinary excretion of a number of electrolytes in healthy horses.

Line 39: 'This information has clinical implications on the short-term effects of hypermagnesemia on calcium-regulating hormones as well as plasma and urine electrolytes.' This needs to be reworded. The clinical implications aren't on the short-term effects of hypermagnesemia. Perhaps, 'clinical implications for the short-term effects of hypermagnesemia' is more appropriate.

As this is an important conclusion of the study, more detail should be provided in the discussion with further explanation of the clinical implications. How important are these clinical implications? What are the potential consequences for the patient? Are the clinical effects likely to be acute or is there potential for chronic alterations in calcium-regulating hormones and plasma and urinary electrolytes?

5. As this study has been performed in healthy horses, the authors should discuss whether similar responses in the parameters of interest may be expected to occur in clinical cases? For example, would similar effects be expected in a horse presenting with hypomagnesemia or a dysrhythmia? If this cannot be ascertained from this study, the authors could still provide some discussion on proposed effects and/or how this could be further investigated.

6. PLOS authors have the option to publish the peer review history of their article (what does this mean?). If published, this will include your full peer review and any attached files.

Reviewer #1: No

Reviewer #2: No

---

## [Author Response · Author response to Decision Letter 0]

19 May 2021

Effects of Intravenous Magnesium Sulfate on Serum Calcium-Regulating Hormones and Plasma and Urinary Electrolytes in Healthy Horses

PONE-D-21-04318

• We appreciate the comments and suggestions by the reviewers to improve this manuscript. Comments are valid and have been addressed.

Reviewer #1: This manuscript is well written, the physiology governing these electrolytes is systematically approached in the introduction and revisited in the discussion. The motivation for this study is clear to the reader; in order to understand how the nefarious use of supplements/drugs in competition horses might result in potential negative clinical implications we first need to understand their effect on physiological status of the horse. I would like to see a better explanation of how Mg has a calming influence on the demeanor of horses, if this is described, with citation(s).

• Thank you for the comments. We have added some information on the potential mechanisms behind the calming effects of Mg in horses. We have previously shown that hypermagnesemia has calming effects/reduces locomotion in horses (Refs #2,3), however, for this manuscript we did not expand on this subject since the goal of this study was specific to hormones and electrolytes and we feel the neurological aspect was beyond the scope of this article. We could provide additional mechanistic information if the reviewer prefers.

Reviewer #2: This paper by Schumacher et al. investigates the effect of experimentally induced hypermagnesemia (using intravenous administration of magnesium sulfate) on blood parathyroid hormone (PTH), calcitonin, ionized calcium, ionized magnesium, sodium, potassium, chloride and their urinary fractional excretion. The authors speculate that the effects of hypermagnesemia on these plasma and urinary variables could have clinical implications and this forms the justification for the research.

Overall, this is a well considered and well conducted study. The manuscript is succinct and well written. The authors provide sound justification/reasoning for conducting the research. The introduction includes relevant information regarding the physiology of magnesium and calcium and the effects of magnesium on calcium regulation which provides appropriate background information for the study. Knowledge of the physiology also supports the hypothesis that there could be effects of hypermagnesemia on plasma and urinary variables. The authors clearly state their hypothesis at the end of the introduction.

The materials and methods section is succinct, yet provides adequate detail on how the study was conducted. The authors clearly explain how the dose of magnesium sulfate used in this study was determined.

The statistical analyses are appropriate and results are clearly presented. The figures provide a clear representation of the results and all results are provided in the accompanying tables.

The discussion provides a succinct summary of the results of the study and the authors clearly discuss their interpretation and provide excellent and detailed reasoning (based on physiological concepts and supported by relevant literature) for the results found. Ideas for research extending beyond this study, including investigating the effects of prolonged hypermagnesemia, are also discussed in the conclusion.

• Thank you for your comments and recognizing the strengths of this study.

There are some minor points listed below which the authors should address before the submission can be accepted for publication.

1. Title: the title needs the addition of 'and' ("Effects of Intravenous Magnesium Sulfate on Calcium-Regulating Hormones and Plasma and Urinary Electrolytes in Healthy Horses') to clarify that the effects on plasma electrolytes etc is of interest, not the effects on plasma.

• The title has been revised as suggested.

2. The research was conducted in 12 horses. Was a sample size calculation performed? This should be discussed in the materials and methods.

• Yes. Additional details were included in the data analysis about power calculation.

3. Line 78: At the end of this sentence, the authors state that the effects of hypermagnesemia could have clinical implications. The authors should expand on this slightly.

• Yes. Additional details were included on potential clinical implications.

4. The authors conclude that the hypermagnesemia alters calcium-regulating hormones, reduces plasma calcium concentrations and increases the urinary excretion of a number of electrolytes in healthy horses.

Line 39: 'This information has clinical implications on the short-term effects of hypermagnesemia on calcium-regulating hormones as well as plasma and urine electrolytes.' This needs to be reworded. The clinical implications aren't on the short-term effects of hypermagnesemia. Perhaps, 'clinical implications for the short-term effects of hypermagnesemia' is more appropriate.

• Valid comment. We rephrase this sentence (lines 39-42).

As this is an important conclusion of the study, more detail should be provided in the discussion with further explanation of the clinical implications. How important are these clinical implications? What are the potential consequences for the patient? Are the clinical effects likely to be acute or is there potential for chronic alterations in calcium-regulating hormones and plasma and urinary electrolytes?

• Additional information on potential clinical implications was included (lines 272-289).

5. As this study has been performed in healthy horses, the authors should discuss whether similar responses in the parameters of interest may be expected to occur in clinical cases? For example, would similar effects be expected in a horse presenting with hypomagnesemia or a dysrhythmia? If this cannot be ascertained from this study, the authors could still provide some discussion on proposed effects and/or how this could be further investigated.

• Information on potential implications of Mg in clinical cases was included (lines 272-289).

---

## [Editor Report · Decision Letter 1]

24 May 2021

Effects of Intravenous Magnesium Sulfate on Serum Calcium-Regulating Hormones and Plasma and Urinary Electrolytes in Healthy Horses

PONE-D-21-04318R1

Dear Dr. Toribio,

We’re pleased to inform you that your manuscript has been judged scientifically suitable for publication and will be formally accepted for publication once it meets all outstanding technical requirements.

Kind regards,

Angel Abuelo, DVM, MRes, MSc, PhD, DABVP (Dairy), DECBHM

Academic Editor

PLOS ONE
---

## [Editor Report · Acceptance letter]

14 Jun 2021

PONE-D-21-04318R1 

Effects of Intravenous Magnesium Sulfate on Serum Calcium-Regulating Hormones and Plasma and Urinary Electrolytes in Healthy Horses 

Dear Dr. Toribio:

I'm pleased to inform you that your manuscript has been deemed suitable for publication in PLOS ONE. Congratulations! Your manuscript is now with our production department. 

Kind regards, 

on behalf of

Dr. Angel Abuelo 

Academic Editor

PLOS ONE